# Pharmacogenetics of *CYP2A6*, *CYP2B6*, and *UGT2B7* in the Context of HIV Treatments in African Populations

**DOI:** 10.3390/jpm12122013

**Published:** 2022-12-05

**Authors:** Graeme R. Ford, Antoinette Niehaus, Fourie Joubert, Michael S. Pepper

**Affiliations:** 1Institute for Cellular and Molecular Medicine, Department of Medical Immunology, University of Pretoria, Pretoria 0002, South Africa; 2Center for Bioinformatics and Computational Biology, Department of Biochemistry, Genetics and Microbiology, University of Pretoria, Pretoria 0002, South Africa

**Keywords:** pharmacogenetics, pharmacogenomic variants, Efavirenz, cyptochrome P-450 enzyme system, precision medicine

## Abstract

Objectives: This study focuses on identifying variations in selected CYP genes related to treatment responses in patients with HIV in African populations by investigating variant characteristics and effects in African cohorts. Design: Cytochrome P450 (CYP) 2A6, 2B6, and Uridine 5’-diphospho-glucuronosyltransferase (UGT) 2B7 allele frequencies were studied using public-domain datasets obtained from the 1000 Genomes Phase 3 project, the African Genome Variation Project (AGVP), and the South African Human Genome Programme (SAHGP). Methods: Variant annotations were performed using self-identified ethnicities to conduct allele frequency analysis in a population-stratification-sensitive manner. The NCBI DB-SNP database was used to identify documented variants and standard frequencies, and the E! Ensembl Variant Effect Predictor tool was used to perform the prediction of possible deleterious variants. Results: A total of 4468 variants were identified across 3676 individuals following pre-filtering. Seventy-one variants were identified at an allelic frequency (1% or more in at least one population), which were predicted to be linked to existing disease associations and, in some cases, linked to drug metabolisms. This list was further studied to identify 23 alleles with disease considerations found at significantly different frequencies in one or more populations. Conclusions: This study describes allele frequencies observed in African populations at significantly different frequencies relative to at least one other reference population and identifies a subset of variants of clinical interest. Despite the inclusion of mixed sequence coverage datasets, the variants identified pose notable avenues for future inquiries. A subset of variants of clinical interest with statistically significant inter-population frequency differences was identified for further inspection, which provides evidence of an African population-specific variant frequency profile. This study highlights the need for additional research and African genetics data given the presence of this unique frequency profile to better facilitate the genetic pre-screening of patients as a standard of practice in HIV care, particularly on the African continent where HIV is highly prevalent.

## 1. Introduction

Efavirenz (EFV) and Nevirapine (NVP) are highly effective non-nucleoside reverse transcriptase inhibitors (NNRTI’s) widely used in combination antiretroviral therapy (cART). In South Africa, they are preferentially used in first-line cART regimens to treat an estimated 7.8 million individuals living with HIV as of 2020, an effective 13% of South Africa’s total population [1]. This is also likely an inaccurate under-representation of the true number of HIV-infected individuals, given previous studies that highlight the effect of stigma on protective sexual behaviours and their effects on patient–doctor relationships [2,3,4,5]. Given the importance of HIV treatments in this context, pharmacogenetic considerations for these HIV treatments should be prioritised to ensure positive treatment outcomes.

The Cytochrome P450 gene family is responsible for metabolising most modern-day pharmaceuticals in first-pass or primary metabolism. EFV and NVP are primarily metabolised by highly polymorphic *CYP2A6*, *CYP2B6*, CYP3A4, and *UGT2B7* enzymes [2,3,4,5]. As a result, unique genetic variation documented within these genes has been found to play a notable role in drug pharmacokinetics [6]. By extension, this genetic variation contributes to inter-individual variations in patient responses, making it an ideal metric to target in the interest of realizing personalized medicine [6]. Various initiatives have attempted to address this integration of pharmacogenetic variations and its effects on prescriptive practices, such as the MedeA Initiative; however, no such initiatives have been identified to be operating in Africa [7,8].

### 1.1. Known Variants

#### 1.1.1. *CYP2A6*

*CYP2A6* has been implicated as a primary metaboliser of EFV and is expressed predominantly in the liver [2,3,4,9,10]. Variant rs28399433, a core variant for the decreased-activity haplotype *CYP2A6*9A*, has been found in all super populations except for African populations [11,12,13]. In contrast, variant rs5031016 has been implicated in three decreased-activity haplotypes but is only common to East Asian populations [11,12,13]. Similarly, variants rs1809810 and rs6413474 have been found mainly in European populations [11,12,13]. Of the two, only rs6413474 has been identified in another population, namely in American populations [11,12,13]. Moreover, rs143731390 has been found only in European and American populations [11,12,13]. Only two SNPs, rs28399454 and rs28399440, were found to occur above the allele frequency (1%) in African populations [11,12,13]. While both SNPs also occur in American populations, rs28399440 also occurs rarely in European and South Asian populations [11,12,13].

#### 1.1.2. *CYP2B6*

*CYP2B6* is involved in the metabolism of both EFV and NVP. Several core SNPs have been identified that occur above the allele frequency; however, rs3745274 and rs2279343 occur most commonly. These are found in African, American, European and East Asian populations [11,12,13]. Only rs3745274 is also found in South Asian populations [11,12,13]. Interestingly, rs2279343 has been found to increase enzyme activity despite its presence in several decreased activity haplotypes [11]. rs3211371 and rs34223104 are also examples of population-specific SNPs that have been linked to decreased activity haplotypes and occur in American populations [11,12,13]. rs3211371 has additionally been found in European populations, while rs34223104 has been found in African populations [11,12,13]. Only two core SNPs above the allele frequency listed for *CYP2B6* are population specific. These are rs28399499, which occur in African populations, and rs34826503, which occurs in South Asian populations [11,12,13].

#### 1.1.3. *CYP3A4*

*CYP3A4* is a highly polymorphic gene expressed in the gastrointestinal tract, liver, and gall bladder and is linked to the metabolism of several modern pharmaceuticals [5,9,10]. Despite this, it is not considered an ideal candidate for pharmacogenetic interventions due to poor genotype–phenotype correlations.

#### 1.1.4. *UGT2B7*

Implicated in the metabolism of EFV, *UGT2B7* is primarily expressed in the liver, gallbladder, kidney and urinary bladder, the pancreas and the gastrointestinal tract [2,4,9,14]. While not currently curated by PharmVar, haplotypes and associated SNPs of the UGT gene-family are maintained by the Canadian Research Chair in Pharmacogenetics [15]. According to this database, however, there are no currently listed, recognised haplotypes of phenotypic effect.

### 1.2. Treatment and Dosage Guidelines

South Africa’s current standard treatment guidelines and essential medicines list indicate that dosage adjustments are not required for EFV [8]. To date, the only pharmacokinetic interaction acknowledged is the EFV concentration increase associated with slow genetic metabolisers due to inhibition by Isoniazid (INH) [8]. It has been established that variations within the *CYP2A6*, *CYP2B6* and *UGT2B7* genes can influence patient responses, and co-prescription with Rifampicin (RMP) and INH may adversely affect NVP concentrations via CYP3A4 and 2B6 gene regulation [8]. As such, these genes have been extensively studied as ideal targets in the field of pharmacogenetics, and several guidelines have been published by the Clinical Pharmacogene Implementation Consortium (CPIC) [16]. One major obstacle to implementing these findings and the CPIC guidelines in Africa is the lack of population-sensitive variant analyses to draw upon in a clinical setting [17].

This study aims to address this by exploring variants found in a cohort of African samples. In addition, this study aims to characterise their effects using in silico methods and to explore potentially pharmacogenetically relevant variants.

## 2. Materials and Methods

### 2.1. Data and Ethics

Variant-Call-Format (VCF) data from 1992 African, 347 American, 503 European, 504 East Asian and 489 South Asian self-identified individuals were included in this study (Ethics Reference Number: NAS111/2019), as summarised in Appendix I-A. Permission was obtained from the African Genome Variation Project (AGVP) and the Southern African Human Genome Programm’s (SAHGP) data access committees in addition to public-access 1000 Genomes Phase 3 data. Low-resolution whole-genome sequencing (WGS), deep exome sequencing and dense microarray genotyping data were obtained from the 1000 Genomes Phase 3 project, high-resolution WGS data from SAHGP and low-resolution chip-based targeted sequence data were obtained from AGVP.

### 2.2. Analysis and Pipeline Development

The analysis was conducted using Snakemake, a python-based workflow management system developed for scalable and reproducible data analyses [18].

#### Data Preparation

Format adherence and data validation were first performed using the GATK-4.0.12.0 tool SelectVariants and FixVcfHeader [19]. Datasets were then lifted over from the hg19 to GRCh38 human genome coordinate systems using the Picard LiftoverVcf tool and chain files obtained from the University of California Santa Cruz (UCSC). During this process, all variants were stripped of rsIDs and renamed using nomenclature (e.g., rs145308399 would become chr19: 40849872C-T) using PLINK-2.0 [20]. The BCFtools software was then used to merge each dataset into a single multi-sample file to facilitate downstream analyses [21]. The resulting data were then annotated against dbSNP (build 146 as provided in the GATK Resource Bundle version 2.8) using BCFtools [21]. The annotated data were then trimmed to regions of interest (see Table 1) and filtered to remove samples with less than a 100% call rate using PLINK-2.0 [20,22].

## 3. Analysis

A population stratified frequency analysis was conducted using PLINK-2.0 [20,22]. A post hoc significance test was then conducted to identify frequency variations of significance (α = 0.05) between African and non-African populations for each variant. The Fisher’s exact test, implemented in the Python SciPy package, was used to calculate a two-tailed *p*-value and unconditional maximum likelihood estimate odds ratio [23]. This test was run using a contingency table to test the association between population grouping (using African as reference) and the allele found (reference or alternate allele) for each variant identified. Bonferroni corrections were performed per gene per population.

The E! Ensembl REST API was used to perform phenotype effect prediction and the meta-analysis for each variant identified using in-house python scripts [24]. The SIFT and PolyPhen algorithms were used to predict deleterious variants, and known variants from the literature were identified and marked. A CONsensus DELeteriousness (CONDEL) weighted average score was calculated to provide an improved assessment of the above scores [25].

The prediction results were then subset using a degree-of-support approach. Singletons represented variants with only one out of three deleterious predictions; doubletons represented variants with two out of three harmful predictions and consensus predictions between all algorithms.

## 4. Results

### 4.1. Variant and Population Partitions Characterisation

To better understand population-based *CYP2A6*, *CYP2B6* and *UGT2B7* variant profiles, a baseline was established by exploring a series of descriptive metrics for each variant. Further subsets were then isolated to identify variants of interest.

The distribution of variant types was first investigated and visualised in Figure 1 to provide insight into variant type compositions between genes. Most variants identified were classed as intronic variants. Missense variants were the second largest variant type found in *CYP2A6* in contrast to *CYP2B6* and *UGT2B7*, where upstream gene variants represented the second-largest variant type.

Following variant type characterisations, the variant frequency was investigated to understand per-gene and per-population variations. A total of 4471 variants were identified, as illustrated in Table 2. Of the total pool of variants identified in this study, 2660 (59.3%) were found to be present in AFR, 1599 (35.8%) in AMR, 1415 (31.6%) in EAS, 1429 (32.0%) in EUR and 1571 (35.1%) in SAS (Figure 2).

Inter-population differences in variant presence was explored via a partition analysis. UpSetPlots were generated to visualise inter-population profiles by segmenting populations into discrete segments based on variant presence/absence (Figure 2). African-containing partitions constituted the highest proportion of total identified alleles (*CYP2A6*: 87.9%; *CYP2B6*: 92.5%; and *UGT2B7*: 92.7%), followed by American-containing partitions (*CYP2A6*: 77.4%; *CYP2B6*: 87.9%; and *UGT2B7*: 87.3%). South-Asian-containing allele partitions constituted the third-largest cross-section of alleles in *CYP2A6* and *UGT2B7* (69.4% and 71.7%, respectively), while European-containing partitions were identified as the third-largest *CYP2B6* partition (74.4%). East-Asian-containing allele partitions were consistently the smallest for all three genes studied (*CYP2A6*: 56.5%; *CYP2B6*: 58.5%; and *UGT2B7*: 62.8%).

The largest partition composition type across all three genes was identified as alleles shared amongst all populations, ranging in size from a minimum of 45.2% (*CYP2A6*) to a maximum of 57.9% (*UGT2B7*). The African-American partition represented the second largest non-universal multi-population partition for all three genes (*CYP2A6*: 12.1%; *CYP2B6*: 14.1%; and *UGT2B7*: 17.6%).

Single-population partitions, denoting variants unique to that population only, were considered to determine the degree of population-unique variations captured in this study. Of the single-population partitions identified, only African, East Asian and South Asian partitions were identified across all three genes. Only *CYP2B6* displayed a European-only partition, while an American-only partition was only found in *UGT2B7*. The African partition was identified as the largest single-population partition in all three genes (*CYP2A6*: 12.1%, *CYP2B6*: 4.9% and *UGT2B7*: 6.9%, as shown in Figure 2), highlighting a higher degree of variation in the African population studied compared to other populations. In comparison, the East-Asian partition was found to be the 10th largest partition for *CYP2A6* (1.6%) and the 6th largest for *CYP2B6* (2.9%) and *UGT2B7* (1.4%). The South Asian allele partition was found to be the 6th largest for *CYP2A6* (2.4%), 10th largest for *CYP2B6* (1.2%) and the 5th largest for *UGT2B7* (1.6%). Finally, the *CYP2B6* European partition was identified as the 13th largest partition at 0.6%, while the American-only *UGT2B7* partition was found to be the 14th largest partition at 0.4%.

### 4.2. Fisher’s Exact Test

A Fisher’s Exact test with Bonferroni correction was implemented to investigate subsets of variants with statistical support (α = 0.05) to indicate significantly different frequencies in African populations compared to the studied populations.

A total of 100 (24.21%) American, 111 (26.88%) European, 117 (28.33%) East Asian and 125 (30.27%) South Asian *CYP2A6* variants were found at significantly different frequencies. No East Asian variants were identified with significantly different frequencies. Of the significant variants, 72 (72%) American, 77 (69.37%) European, 87 (74.36%) East Asian and 84 (67.2%) South Asian variants were found to show significant odds ratios greater than one, indicating significantly higher frequencies of the alternate allele in African populations. In contrast, 28 (28%) American, 34 (30.63%) European, 30 (25.64%) East Asian and 41 (32.8%) South Asian variants were observed with a significant odds ratio of less than one, indicating significantly lower frequencies of alternate alleles in African populations.

In comparison, 497 (31.34%) American, 674 (42.5%) European, 666 (41.99%) East Asian, and 613 (38.65%) South Asian *CYP2B6* variants were found to have significant Fisher’s exact scores. When investigating these, 435 American (87.53%), 591 (87.69%) European, 592 (88.89%) East Asian and 523 (85.32%) South Asian variants showed significant odds ratios greater than one, indicating a higher alternate allele frequency in African populations. In comparison, 62 (12.47%) American, 83 (12.31%) European, 74 (11.11%) East Asian and 90 (14.68%) South Asian variants showed significant odds ratios of less than one, indicating a lower alternate allele frequency in African populations.

Finally, a total of 671 (27.18%) American, 754 (30.54%) European, 789 (31.96%) East Asian and 777 (31.47%) South Asian *UGT2B7* variants were found to have significant Fishers-Exact scores. Of these significant variants, 551 (82.11%) American, 644 (85.41%) European, 660 (83.65%) East Asian and 643 (82.75%) South Asian variants were identified with significant odds ratios greater than one, indicating higher alternate allele frequencies in African populations; 120 (17.88%) American, 110 (14.59%) European, 129 (16.35%) East Asian and 134 (17.25%) South Asian variants were identified with significant odds ratios of less than one, indicating lower alternate allele frequencies in African populations.

### 4.3. Variant Effect Prediction

#### 4.3.1. Variants with Known Phenotype Associations

Following variant effect prediction (VEP), a post hoc CONDEL score was implemented. Various variant subsets of possible medical interest were then identified for cross-comparisons with frequency data.

A total of 371 *CYP2A6*, 1305 *CYP2B6* and 34 *UGT2B7* variants were linked to phenotypic responses by the E! Ensembl VEP database [24]. Further analyses revealed 109 *CYP2A6*, 289 *CYP2B6* and 31 *UGT2B7* variants at an allelic frequency with identified VEP phenotypes (Appendix A).

A total of 94 *CYP2A6*, 258 *CYP2B6* and 30 *UGT2B7* alleles were identified with known disease phenotype associations and significantly different frequencies compared to African populations. All 258 *CYP2B6* variants were found to be associated with the poor metabolism of Efavirenz and potential central nervous system toxicity (Appendix A). In addition, alleles rs3745274 and rs28399499 were also linked to Nevirapine metabolism and toxicity. Additional *CYP2B6* phenotype co-associations were also found for the DDT metabolism (3), polychlorinated biphenyl levels (2), HDL cholesterol levels in HIV infection (1), smoking behaviour (1) and midgestational circulating levels of organochlorine pesticides (1) and polybrominated diphenyl ethers (1) respectively. None of the *CYP2A6* or *UGT2B7* alleles identified were associated with Efavirens or Nevirapine.

In order to assess the impact of each variant in this study, both variants with CONDEL predictions and variants with known variant phenotype associations were considered. In order to investigate deleterious variants identified by this method, variants at an allelic frequency in at least one population and with a frequency significantly different from at least one other population were considered.

Twelve variants were identified in *CYP2A6*, nine alleles in *CYP2B6* and two alleles in *UGT2B7* that satisfied the above conditions (Table 3), which were further examined relative to African populations. A total of 6 alleles (4 *CYP2A6* and 2 *CYP2B6* alleles) were found to occur significantly more often while 13 alleles (7 *CYP2A6*, 5 *CYP2B6* and 1 *UGT2B7* allele) were found to occur significantly less often in American populations. When examining European populations, 6 alleles (4 *CYP2A6* and 2 *CYP2B6* alleles) were found to occur at significantly higher frequencies while 14 (6 *CYP2A6*, 7 *CYP2B6* and 1 *UGT2B7* allele) were found to occur significantly less often. Two alleles (one *CYP2A6* and one *CYP2B6* allele) were found to occur significantly more often in East Asian populations, while fourteen alleles were found to occur significantly less often. Finally, seven alleles (five *CYP2A6*, one *CYP2B6* and one *UGT2B7* allele) were found to occur significantly.

Of the *CYP2A6* alleles observed above, all twelve alleles were associated with lung cancer (alveolar cell carcinomas), susceptibility to tobacco addiction, Letrozole toxicity and coumarin resistance. rs1801272 was additionally associated with poor nicotine metabolism and, consequently, was recorded to be associated with smoking behavior as well as warfarin response and C-reactive protein levels. Importantly, rs1801272 was found to be significantly less frequent in African populations (0.08%) compared to American (0.72%), European (3.38%) and South Asian populations (0.61%) and was not found at allelic frequency (0.08%), contrary to European populations (3.38%). In total, five alleles (rs6413474, rs1809810, rs199916117, rs1801272 and rs28399435) with disease associations were found to occur at significantly lower frequencies in the populations studied, while six (rs28399463, rs28399454, rs56256500, rs28399440, rs72549435 and rs72549432) were found to occur at significantly higher frequencies. rs145308399 was observed at higher frequencies in African populations compared to American populations but lower frequencies when compared to South Asian populations. Importantly, rs199916117 and rs6413474 were not observed in African populations, while rs56256500, rs28399440 and rs28399440 were not observed in any other population and were all observed at allelic frequencies in Africans. This highlights rs56256500, rs28399440 and rs28399440 as potential targets for population-specific studies with disease-centric study focus.

When considering *CYP2B6* alleles, all nine alleles were associated with poor Efavirenz metabolism resulting in central nervous system toxicity. Interestingly, both rs3745274 and rs28399499 were additionally associated with Nevirapine metabolism and toxicity, respectively. Seven of the alleles were found to occur significantly more frequently in African populations when compared to comparison populations, while rs34883432 was found to be absent in African populations compared to American (1.3%) and European populations (0.7%) and rs3211371 was found to occur at significantly higher frequencies in African populations (1.21%) compared to East Asian populations (0.3%) but lower when compared to American (7.21%), European (11.23%) and South Asian (8.9%) populations. Importantly, rs33980385, rs33926104, rs34284776 and rs139029625 were not found at allelic frequencies in any other population aside from African populations, which also implicates these alleles in African-sensitive disease research.

Finally, of the two *UGT2B7* variants, both were associated with tramadol responses. rs12233719 was observed at a significantly lower frequency in African populations (0.08%) compared to East and South Asian populations, while rs7439366 was found to occur at significantly higher frequencies in African populations (77.49%) when compared to American (68.01%), European (51.49%), East (72.52%) and South Asian (60.12%) populations. rs7439366 was found to occur in a notable portion of African populations, which was larger than any other population, making this allele exceptionally common.

#### 4.3.2. Novel Variants and Their Potential Pathogenicity

Potentially novel variants were identified after performing variant annotations using the dbSNP build 146 as well as the Ensembl REST API. A total of 2 *CYP2A6*, 6 *CYP2B6* and 18 *UGT2B7* variants were identified (Table 4). Of these, 1 was a downstream variant, 21 were intronic variants, 2 were 3’ UTR variants and 2 were unclassified. None of these identified novel variants were linked to corresponding VEP entries in the E! Ensembl database [24]. As such, SIFT and PolyPhen scores were unavailable, and subsequently, a CONDEL score was not possible for these variants.

When considering the CONDEL predictor of variant effect and variants of allelic status, 4 alleles were classified as deleterious and 12 alleles were classified as neutral for *CYP2A6*. Seven neutral and two deleterious variants were found for *CYP2B6* and four neutral and two delirious alleles were found for *UGT2B7*. Of these alleles, all alleles were matched to existing variants in the E! Ensembl VEP database.

## 5. Discussion

The main objective of this study was to compare variant frequencies and their impacts across population groups and to ascertain the possible implications for clinical prescriptive practices.

### 5.1. Variant Frequency

An analysis of variant frequency rates amongst populations (Appendix A) revealed a correlation coefficient of less than one across all populations for *CYP2A6* with an average coefficient of 0.88 (S.D. ± 0.08). *CYP2B6* displayed coefficients above 1 for East Asian and European populations and less than one for South Asian and American populations with an average of 1.08 (S.D. ± 0.13). In contrast, *UGT2B7* displayed a coefficient of above one across all populations with an average of 1.30 (S.D. ± 0.16). Interestingly, the standard deviation of coefficients was found to follow a linear relationship relative to gene size with an R^2^ value of 96.21% (Appendix A).

The frequency partition analysis revealed the presence of a sizeable African-unique allele partition for each gene studied. Furthermore, the African population was also represented in all of the largest three multi-population partitions across all three genes. This indicates partial cross-population sharing in addition to a high degree of inherent genetic diversity. Interestingly, of the studied alleles that were identified, few were characterizable with CONDEL variant effect predictions. These findings indicate a possible deficit in using in silico pre-calculated databases, which rely on historical data and understanding the genome. One possible solution to this is the implementation of additional, more advanced phenotype prediction methods, which should be considered in future.

Interestingly, only a single partition of variants uniquely attributed to American samples was identified in *UGT2B7*. This is despite a large pool of variants shared amongst many other populations, the most notable of which is the African-American partition, which was found to be the second largest non-population-spesific partition in all three genes. These findings show a relatively low degree of population-specific variation and a high degree of shared variation in the American populations studied with the largest association being African-American variants. While African-American-labelled samples were removed before analysis to control for admixture bias and prevent over or under-representation of variation associated with admixture, these results would indicate the presence of additional unrecorded potential admixture.

Odds-ratios were investigated to describe significant (α = 0.05) allele frequency (1%) differences between populations. *UGT2B7* showed a much greater number of variants with odds ratios greater than or equal to 1 than those with ratios less than 1 (Appendix A) for all populations compared. This indicates these disease-associated alleles occur more frequently in African populations compared to each comparison population. *CYP2B6* followed this trend for all populations except the South Asian population, where the majority of disease-associated alleles were found to occur less often in African populations than in South Asian populations. In contrast to *CYP2B6*, most *CYP2A6* disease-associated alleles except those in the African-American partition were found to occur less often in African populations than in each comparison population. These variable odds ratio profiles for each gene and population indicate the presence of distinct frequency profiles and phenotype relations between these populations and genes.

One point of interest was the high degree of disease-associated alleles with significant frequency differences noted in *CYP2B6* compared to *CYP2A6* and *UGT2B7*. This difference was unexpected given the size of the CYP genes relative to that of *UGT2B7*, in contrast to the earlier linear trend of gene length and average variant frequency deviation noted in Appendix A.

### 5.2. Variant Impact

When reviewing variants identified in Table 3, ten CYP2A6, four *CYP2B6* and all *UGT2B7* alleles of practical note were identified in the PharmVar and PharmGKB databases with registered clinical annotations [11,26]. Variants identified in *CYP2A6* have not all been associated with a CPIC clinical allele function. This has been provided in the form of manual revision by the PharmVar maintainers in the form of manual classifications reviewed by a panel [27]. Furthermore, all *CYP2A6* variants were associated with a gene-duplication haplotype, and CYP2A6*1x2 is associated with an increased metabolism [11].

In review, a total of 13 variants across all genes studied were found at significantly higher frequencies in African populations than in a comparison population, while 6 variants were found at significantly lower frequencies (Table 3). Interestingly, the variant rs28399454 was identified in connection with *CYP2A6*17*, a reduced-function haplotype known from the literature [11,12,13,27]. This variant was found at allelic statuses in African populations at significantly higher rates (11.9%) compared to American (0.58%) and East Asian (0.1%) populations. When compared against the ALFA project, it was observed at a higher frequency in our dataset at 11.9% while ALFA records it at 9.7% [12,13]. Additionally, rs1801272 and rs28399440 were also identified with known haplotype associations, namely the *CYP2A6*2* and **9* haplotypes, respectively. Both of these haplotypes were associated with decreased gene functions; however, rs1801272 was not found at an allelic frequency in African populations (0.08%), while rs28399440 was (1.81%), indicating increased potential *CYP2A6*9* haplotype prevalence compared to *CYP2A6*2*. Of these, rs28399440 was found at a slightly higher frequency of 1.81% compared to the ALFA project at 1.12% [12,13].

Three *CYP2B6* alleles were identified, namely rs3745274, rs139029625 and rs28399499, which were associated with loss-of-function haplotypes and found to be significantly more common in African populations. In addition to the loss-of-function *CYP2B6*37* and **38* haplotypes, rs3745274 was also associated with the reduced function haplotypes *CYP2B6*6*, **7*, **9*, **19*, **20*, **26*, **34* and **36* haplotypes, all of which were assigned reduced function phenotypes [11]. This variant was found at a frequency of 37.95% in African populations, making it an extremely prolific variant compared to European and East Asian populations and slightly more common than is expected from the ALFA project, which has recorded it at 36.72%. rs139029625 and rs139029625 in contrast were associated with loss-of-function haplotypes *CYP2B6*35* and **18*, respectively. Importantly, rs139029625 transversion mutations from Guanine to Cytosine were only identified in African populations at an allelic frequency of 1.21%, which is in contrast to recorded ALFA frequencies that place the mutation at 0.06%, while rs28399499 was only identified in African (8.56%) and American (1.01%) populations [12,13]. Importantly, rs28399499 was observed at higher frequencies in African populations than the expected ALFA frequencies of 6.84% [12,13]. In contrast, rs3211371, which was associated with *CYP2B6*7* and **34*, both reduced function haplotypes, was found to occur at significantly lower frequencies in African populations (1.21%) compared to all comparison populations (AMR: 7.21%; EUR: 11.23%; and SAS: 8.9%), except East Asian populations (0.3%). This was slightly lower than expected from the ALFA project recorded African population frequency of 1.45% [11,13]. This highlights rs3745274, rs139029625 and rs28399499 as ideal targets for African-specific variations andthey are ideal targets for routine disease screening, but this also shows several differences relative to expected frequencies based on the literature.

Of the *UGT2B7* variants, none were linked to known disease haplotypes; however, rs12233719 was found at significantly lower frequencies in African populations compared to East and South Asian populations and higher than the expected African frequency of 0.01% from the ALPHA project [11,13]. Interestingly, rs12233719 was also found to be slightly more common in European populations at 3.98%, higher than the expected 3.54%, in East Asian populations at 13.19% compared to the ALFA frequency of 15.30% as well as in South Asian populations at 0.72% with ALFA frequency of 0.3% [12,13]. rs7439366 was also found at higher frequencies than expected at 77.49% compared to the expected frequency of 67.94% [12,13]. Interestingly, rs7439366 was also found at higher frequencies in the comparison populations than expected.

## 6. Conclusions

The inclusion of additional African samples showed a high degree of genetic diversity present in these populations. A large degree of shared variation was also identified, although a notable number of subsets have only been identified in some populations. Furthermore, there is statistical support to indicate that these subsets of alleles occurred more/less often in African populations than previously studied populations, indicating the presence of a unique variant profile. Furthermore, alleles have been identified in African populations, which were not observed at allelic frequency in any other population in CYP2A6, while alleles in comparison populations were found to be absent in African populations in *CYP2A6* and *CYP2B6*.

Many variants identified in this study have also been linked to previously documented variants (Appendix A). Several variants were linked to known Nevirapine and Efavirenz disease phenotypes and significantly variable inter-population frequencies. Some of these variants identified have been implicated in significant frequency differences between African and comparison populations, where some were absent in African populations while some were only observed in African populations. These findings pose several significant treatment questions around pharmacogenetic dynamics in clinical protocol and prescriptive practices. A trend can be observed in this study’s findings showing an increase in expected frequencies when compared to the previously recorded literature. This poses important implications for medical treatment and the occurrence of some of the associated deleterious haplotypes. High-quality representative datasets would be incredibly valuable in addressing the need for the high-resolution characterisation of the occurrence and presence of disease-associated variation in African populations and essential to realising routine pharmacogenetic testing and associated dosage adjustments.

### Limitations of the Study

One notable limitation of this study is the need for a comprehensive Admixture analysis. This study used previous findings and knowledge to highlight and exclude known admixed populations such as African-American populations. Despite this, a large African-American partition was still identified, implicating potential unrecorded inter-population gene transfer. Further studies should include admixture analysis and associated sample exclusion to refine the results. In addition to this, the E! Ensembls VEP database provided limited insight into novel variants. This is a cause of concern given the high number of population-specific variations observed [24]. African populations for which high-quality full-sequence data are not abundant or well-studied are especially impacted by this, highlighting a need for more comprehensive in silico variant effect assessment methods.

Despite these limitations, our exploration of population variant profiles highlights the need to standardize the status of known pharmacogenetic variants in African populations. This study and its findings draw attention to the need for population-specific datasets when considering personalized medicine applications, especially when considering currently available datasets that do not accurately represent genetic diversity in African populations. Contingent to this, emphasis should also be placed on re-evaluating current prescriptive practice in a pharmacotype-sensitive manner.

## Figures and Tables

**Figure 1 jpm-12-02013-f001:**
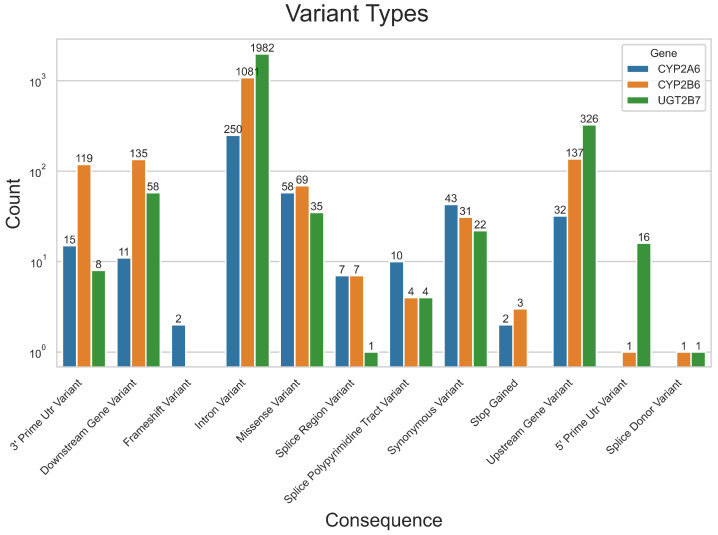
Summary of variations observed in *CYP2A6*, *CYP2B6* and *UGT2B7*, categorized by variation types.

**Figure 2 jpm-12-02013-f002:**
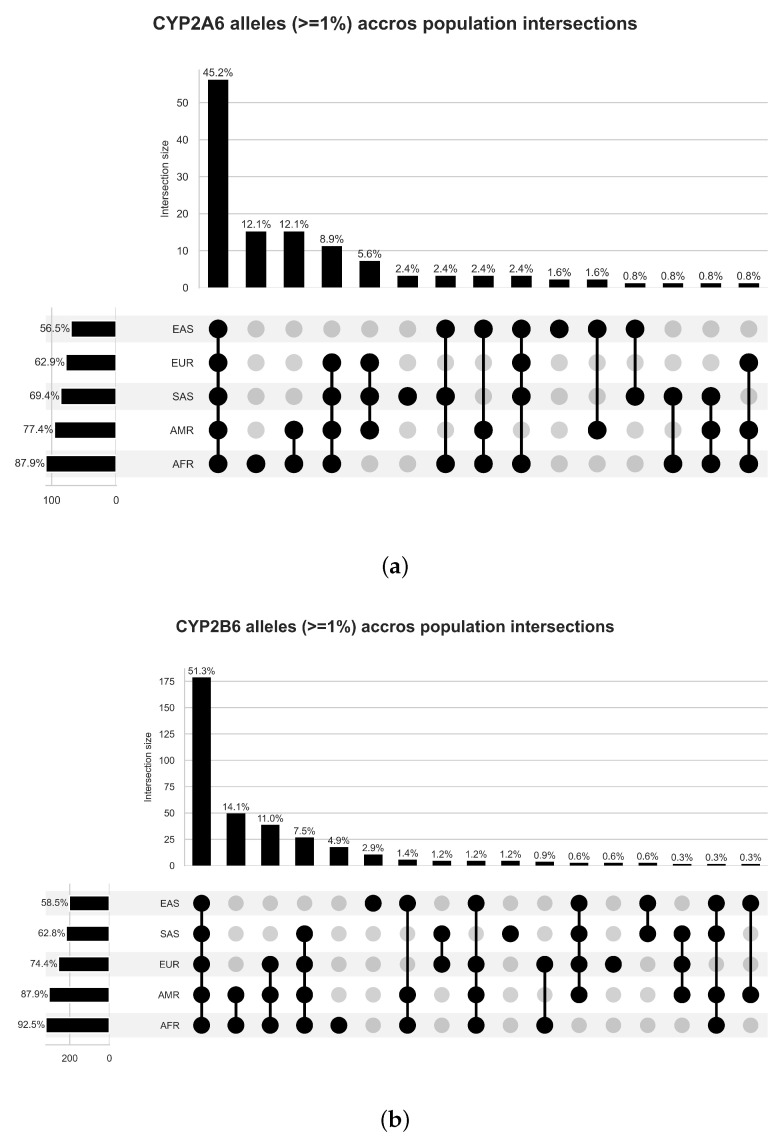
Intersection plots of alleles (>1% Freq) across populations for (**a**) *CYP2A6*, (**b**) *CYP2B6* and (**c**) *UGT2B7*. Partition composition (lower section) is indicated using a series of lines and dots to indicate which populations are included in each partition. In contrast, the upper bar graph indicates partition size (number of variants that satisfied population presence/absence conditions for a given partition) relative to other partitions. The far-left lower vertical bar graph indicates the total population size (number of variants) considered relative to other populations.

**Table 1 jpm-12-02013-t001:** Gene start and stop coordinates used in this study.

Gene	Start Coordinates (Source—100 bp)	Stop Coordinates (Source +100 bp)	Region Size
*CYP2A6*	41,349,343 bp (e! and UCSC)	41,356,460 bp (NCBI)	7117 bp
*CYP2B6*	41,497,104 bp (e! and UCSC)	41,524,408 bp (NCBI)	27,304 bp
*UGT2B7*	69,916,981 bp (e!)	69,978,805 bp (All)	61,624 bp

**Table 2 jpm-12-02013-t002:** A summary of significant alleles for *CYP2A6*, *CYP2B6* and *UGT2B7* ^1^.

Gene	Variants Identified	Alleles (1% or More)	Two-Tailed Significance (α = 0.05)
*CYP2A6*	413	103	159
*CYP2B6*	1586	322	771
*UGT2B7*	2469	736	957

^1^ Allelic status was calculated using a 0.01 frequency cut-off in one of the studied super-populations. Significance was determined using a two-tailed Fisher′s exact test with Bonferroni corrections at a confidence interval of 0.05, where significance was identified in at least one or more of the studied populations.

**Table 3 jpm-12-02013-t003:** Allele haplotype ^1^ and frequency ^2^ summary and relative percentage change for CYP2A6, *CYP2B6* and *UGT2B7* alleles with known CONDEL scores or Disease association.

rsID	Haplotypes^(pheno.)^	AFR ^Δ vs. other^	AMR ^Δ vs. AFR^	EUR ^Δ vs. AFR^	EAS ^Δ vs. AFR^	SAS ^Δ vs. AFR^
rs6413474	**1*, **1*×2^↑^, **21*^†^	0% ^↓^	0.58% ^↑^	1.19% ^↑^	-	0.82% ^↑^
rs28399463	**1*, **1*×2^↑^, **28*^†^, **44*^†^	2.65% ^↑^	0.29% ^↓^	0% ^↓^	0% ^↓^	0.1% ^↓^
rs1809810	**1*, **1*× 2^↑^, **18*^†^, **19*^†^	0.61% ^↓^	1.59% ^↑^	1.59% ^↑^	-	2.25% ^↑^
rs28399454	**1*, **1*× 2^↑^, **17*^↓^	11.9% ^↑^	0.58% ^↓^	0% ^↓^	0.1% ^↓^	0% ^↓^
rs56256500	**1*, **1*× 2^↑^, **16*^†^, **23*^†^	1.97% ^↑^	0% ^↓^	0% ^↓^	0% ^↓^	0% ^↓^
rs199916117	-	0% ^↓^	-	-	1.49% ^↑^	-
rs1801272	**1*, **1*×2^↑^, **2*^↓^	0.08% ^↓^	0.72% ^↑^	3.38% ^↑^	-	0.61% ^↑^
rs28399440	**1*, **1*×2^↑^, **9*^↓^, **13*^†^, **15*^†^, **50*^†^	1.81% ^↑^	0% ^↓^	0% ^↓^	0% ^↓^	0% ^↓^
rs72549435	**1*, **1*× 2^↑^, **24*^†^, **49*^†^	1.06% ^↑^	0.14% ^↓^	0% ^↓^	0.1% ^↓^	0% ^↓^
rs145308399	-	0.15% ^↕^	0% ^↓^	-	-	2.15% ^↑^
rs28399435	**1*, **1*×2^↑^, **14*^†^	0.38% ^↓^	1.73% ^↑^	3.28% ^↑^	-	2.25% ^↑^
rs72549432	**1*, **1*× 2^↑^, **31*^†^	1.13% ^↑^	0% ^↓^	0% ^↓^	0% ^↓^	0% ^↓^
rs34883432	**1*, **10*^‡^	0% ^↓^	1.3% ^↑^	0.7% ^↑^	-	-
rs8192709	**1*, **2*, **10*^‡^	4.3% ^↑^	0.29% ^↓^	6.26% ^↓^	-	-
rs33980385	**1*, **17*	1.29% ^↑^	-	0% ^↓^	0% ^↓^	0% ^↓^
rs33926104	**1*, **17*	1.29% ^↑^	0.29% ^↓^	0% ^↓^	0% ^↓^	0% ^↓^
rs34284776	**1*, **17*	1.29% ^↑^	0.29% ^↓^	0% ^↓^	0% ^↓^	0% ^↓^
rs3745274	**1*, **6*^↓^, **7*^↓^, **9*^↓^, **13*^⊘^, **19*^↓^, **20*^↓^, **26*^↓^, **34*^↓^, **36*^↓^, **37*^⊘^, **38*^⊘^	37.95% ^↑^	-	23.56% ^↓^	21.53% ^↓^	-
rs139029625	**1*, **35*^⊘^	1.21% ^↑^	0% ^↓^	0% ^↓^	0% ^↓^	0% ^↓^
rs28399499	**1*, **18*^⊘^	8.56% ^↑^	1.01% ^↓^	0% ^↓^	0% ^↓^	0% ^↓^
rs3211371	**1*, **5*, **7*^↓^, **33*^‡^, **34*^↓^	1.21% ^↕^	7.21% ^↑^	11.23% ^↑^	0.3% ^↓^	8.9% ^↑^
rs12233719	-	0.08% ^↓^	-	-	13.19% ^↑^	0.72% ^↑^
rs7439366	-	77.49% ^↑^	68.01 ^↓^	51.49% ^↓^	72.52% ^↓^	60.12% ^↓^

^1^ Haplotypes and CPIC clinical allele functions were obtained using the PharmVar Database [11]. The clinical allele function for *CYP2A6* has been assigned by a PharmVar expert panel where no CPIC assignment was available. Alleles, which were not linked to a named haplotype or *CYP2A6* variants, are denoted with ‘-’. ^2^ Only variants with a significant frequency difference between African populations and at least one other comparison population were considered. Frequency differences where a . 0.05 are omitted with ‘-’. ^†^
*CYP2A6* haplotypes that have not been assigned a clinical allele function by CPIC or PharmVar. ^‡^ Haplotypes of uncertain function. ^⊘^ Haplotypes associated with loss of gene function. ^↑^, ^↓^ and ^↕^ indicates net difference when compared to another population. African populations have been compared relative to alternate populations (AMR, EUR, EAS and SAS) while alternate populations have been compared relative to the african population, as indicated in each column header

**Table 4 jpm-12-02013-t004:** Identified novel variants ^1^.

Gene	Position	Reference	Alternate	Consequence
*CYP2A6*	40843668	T	C	3’ UTR variant
*CYP2A6*	40848968	G	GA	Intron variant
*CYP2B6*	40992518	T	C	Intron variant
*CYP2B6*	40995794	ATGATATT	A	Intron variant
*CYP2B6*	40996689	T	C	Intron variant
*CYP2B6*	41001709	G	A	Intron variant
*CYP2B6*	41008210	T	TTTG	Intron variant
*CYP2B6*	41017322	C	A	3’ UTR variant
*UGT2B7*	69046035	T	A	Downstream gene variant
*UGT2B7*	69052469	G	GT	*Mapping Failure* *
*UGT2B7*	69053111	A	G	Intron variant
*UGT2B7*	69056859	T	C	Intron variant
*UGT2B7*	69058254	C	T	Intron variant
*UGT2B7*	69059862	C	G	Intron variant
*UGT2B7*	69060704	G	C	Intron variant
*UGT2B7*	69063327	A	AAAAAAGG	Intron variant
*UGT2B7*	69063329	A	AAAAGAAAG	Intron variant
*UGT2B7*	69064070	A	AAG	Intron variant
*UGT2B7*	69064096	G	GAA	Intron variant
*UGT2B7*	69070106	G	C	Intron variant
*UGT2B7*	69078334	C	T	Intron variant
*UGT2B7*	69092102	A	AT	*Mapping Failure* *
*UGT2B7*	69099275	AAAAG	A	Intron variant
*UGT2B7*	69100656	G	A	Intron variant
*UGT2B7*	69104655	G	A	Intron variant
*UGT2B7*	69107691	T	C	Intron variant

^1^ Novel variants were identified by performing variant annotation against dbSNP build 146 as well as crosschecking for known variants via the Ensembl REST API. * Two variants identified did not map to any existing variant entries. These were both UBT2B7 variants at 69052469 and 69092102 and have been marked accordingly

## Data Availability

The in-house pipeline developed in this study is available through GitHub (https://github.com/Tuks-ICMM/Pharmacogenetic-Analysis-Pipeline/releases/tag/v1.0.0-alpha). The datasets generated during this study are available as Appendix A to this article, hosted by the Journal of Precision Medicine.

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
