# Peer review of "Pharmacogenetics of CYP2A6, CYP2B6, and UGT2B7 in the Context of HIV Treatments in African Populations"

_jpm, 2022, doi:10.3390/jpm12122013_

Round 1
Reviewer 1 Report
Brief Summary
Ford et al. embark on a journey to identify genetic variations that affect HIV drug metabolism. Specifically, they focus on cytochrome P450 enzymes CYP2A6, CYP2B6 and UGT2B7. They highlight the genetic variation in population from Africa, aiming to expand the data availability for this continent. Furthermore, they seek to supply clinicians with genetic distributions relevant to HIV drug prescriptions.
Significance
40 years of research has turned HIV from a deadly disease to a chronic life-long infection. Yet rising antiviral resistance, drug interactions, and side effects from long term prescriptions pose a threat to disease management in the clinics. In these settings, it is now more important than ever to invest in personalized medicine approaches. Differential metabolism of HIV drugs in individuals creates an urgent need to develop clinical diagnostic tools, such as identifying the contributing genetic variations. Furthermore, expanding our understanding of the relevant genetic variations of different populations can direct clinical guidelines and center their medical needs, whether common or unique.
Recommendations:
I recommend accepting this paper with minor revisions for publication at the JPM Journal. I am listing below minor suggestions for clarifying details described in this commentary. Of concern, is purposefully excluding African-American data from analysis, which should be included back.
Comments:
General:
- Please make sure to add a reference to all figures in the text (such as for figure 1).
- Please make sure all the literature references are listed.
- Please elaborate and add to introduction how the gene variation changes the drug prescriptions, with references.
- Please refer to not having South American population in your study.
- No references in discussion and summery. Please add discussion and refer to literature.
- Please make sure to reference all tools and programs used in this paper.
- Please make sure all subtitles have the same format. Such as Line 115: “Data preparation” is not italicized as the rest of the subtitles. Additional suggestion could be to number them.
Line 19 “which also predicted deleterious consequences”. Please expand by adding a few words, such as: “which also predicted deleterious consequence to drug metabolism”.
Line 21: “Conclusions: This study describes allele frequencies in several African sub-populations and identifies previously documented and potentially novel variants of clinical relevance. Despite the mixed sequence coverage, the variants identified pose notable avenues of future inquiry. This study illustrates the need to perform genetic pre-screening of patients as a standard of practice in HIV care, particularly on the African continent where HIV is highly prevalent.” Is this the right conclusion? How about the benefit for the African population? What is the significance? Why study the variation of cytochrome P450? Please elaborate on these questions in the abstract.
Line 34: “This is also likely not an accurate representation of the true number of HIV-infected individuals, given previous studies which highlight the effect of stigma on protective sexual behaviours and their effects on biomedical relationships [2- 5].” Please clarify that the actual numbers are estimated to be higher/underreported.
Lines 125, 156, 165, 167, 189, 209: ”Error! Reference source not found.” Please notice missing references and fix that. If those are references to a figure/table, you could manually insert the correct figure/table number.
Line 142: “E! Ensembl Variant Effect Predictor tool” please add reference to tool. Such as: “McLaren, W., Gil, L., Hunt, S. E., Riat, H. S., Ritchie, G. R. S., Thormann, A., & Cunningham, F. (2016). The ensembl variant effect predictor. Genome Biology, 17(1), 122. https://doi.org/10.1186/s13059-016-0974-4. Or find more instructions here: https://uswest.ensembl.org/info/about/publications.html .
Line 145 “CONDEL”. Please add full acronym at first appearance.
Line 162: “Figure 1. Summary of variant types by the gene.” Please elaborate, for example: “The variation in CYP2A6, CYP2B6 and UGT2B7 genes presented by variation type”.
Line 245: “VEP”. Please add full acronym at first appearance.
Line 253: “All 25 variants identified were associated with the poor metabolism of Efavirenz and potential central nervous system toxicity.” Please add a table or a figure listing those variations, and refer to it in the text.
Line 255:” rs139818840” please describe the variation, here or in the discussion…. Is this table 3?
Line 363: ”to highlight and exclude known admixed populations such as African-American populations”. Why did you exclude the African American population? Are they not included in the American data? Please add this data back into the American data and reanalyze. You can dedicate a section to elaborate on this population genetic variability, in addition to including it in the American data section.
Author Response
Dear Reviewer,
Many thanks for your revisions. Following re-analysis to include additional data which was erroneously not initially included, the manuscript has been reviewed and undergone non-trivial results changes internally in tandem with the provided revision points. Please find below a list of the changes made in relation to your comments, organised as received:
A note on manuscript formatting issues:
When reviewing the provided comments, it was found that the majority of the provided revisions pertaining to formatting guidelines were a result of Microsoft Word formatting having gone awry and suspected issues of the manuscript during transit. To rectify this, the manuscript has since been rendered into a LaTeX document using the approved MDPI LaTeX document template.
General
- Please make sure to add a reference to all figures in the text (such as for figure 1).
Figure references have been corrected using BibTex LaTeX typesetting and referencing.
- Please make sure all the literature references are listed.
Literature references have been reviewed.
- Please elaborate and add to introduction how the gene variation changes the drug prescriptions, with references.
At the time of writing, one genotype-guided dosage prescription and dosage adjustment initative was identified. The MedeA Initiative in Spain are actively pursuing the integration of pharmacogenetics data when deciding drug prescription and dosage. These references have been added to the introduction [LINE ]. Measuring the impact of genetic polymorphisms on drug clearance for efavirenz in African Populations, however, requires a preliminary understanding of the population-specific variation in question. As a result, genotype-guided dosage adjustment is difficult in understudied populations such as the African populations where variant characterization is still taking place.
- Please refer to not having South American population in your study.
At the time of analysis and data collection and the issue of ethical clearance, a dataset representative of South American populations was not identified. Additional datasets have since been identified including the HGDP and will be included in any future studies on this topic.
- No references in discussion and summary. Please add discussion and refer to literature.
The Discussion has been expanded and references have been included.
- Please make sure to reference all tools and programs used in this paper.
References for all tool used has been reviewed.
- Please make sure all subtitles have the same format. Such as Line 115: “Data preparation” is not italicized as the rest of the subtitles. Additional suggestion could be to number them.
Formatting has been corrected by converting the manuscript into LaTeX document format using the approved MDPI template.
Line-wise revisions
- “which also predicted deleterious consequences”. Please expand by adding a few words, such as: “which also predicted deleterious consequence to drug metabolism”.
The abstract has been reviewed and its contents have been expanded. This has been included.
- “Conclusions: This study describes allele frequencies in several African sub-populations and identifies previously documented and potentially novel variants of clinical relevance. Despite the mixed sequence coverage, the variants identified pose notable avenues of future inquiry. This study illustrates the need to perform genetic pre-screening of patients as a standard of practice in HIV care, particularly on the African continent where HIV is highly prevalent.” Is this the right conclusion? How about the benefit for the African population? What is the significance? Why study the variation of cytochrome P450? Please elaborate on these questions in the abstract.
The abstract conclusion has been reviewed and the conclusion has been re-written.
- “This is also likely not an accurate representation of the true number of HIV-infected individuals, given previous studies which highlight the effect of stigma on protective sexual behaviours and their effects on biomedical relationships [2- 5].” Please clarify that the actual numbers are estimated to be higher/underreported.
This has been clarified. The actual numbers are expected to be much higher and largely underreported due to, amongst other factors, social stigmas negatively impacting the discovery and treatment of HIV infections.
- ”Error! Reference source not found.” Please notice missing references and fix that. If those are references to a figure/table, you could manually insert the correct figure/table number.
[LINES 125, 156, 165, 167, 189, 209] Referencing has been corrected. This was a result of the malformed Microsoft Word formatted manuscript initially submitted.
- “E! Ensembl Variant Effect Predictor tool” please add reference to tool. Such as: “McLaren, W., Gil, L., Hunt, S. E., Riat, H. S., Ritchie, G. R. S., Thormann, A., & Cunningham, F. (2016). The ensembl variant effect predictor. Genome Biology, 17(1), 122. https://doi.org/10.1186/s13059-016-0974-4. Or find more instructions here: https://uswest.ensembl.org/info/about/publications.html .
[LINE 142] The requisite reference has been placed for the E! Ensemble VEP API.
- “CONDEL”. Please add full acronym at first appearance.
[LINE 145 ->] The CONDEL acronym has been properly introduced.
- “Figure 1. Summary of variant types by the gene.” Please elaborate, for example: “The variation in CYP2A6, CYP2B6 and UGT2B7 genes presented by variation type”.
[FIGURE 1] The title of Figure 1 has been amended to "Summary of variation observed in CYP2A6, CYP2B6 and UGT2B7, categorized by variation type."
- “All 25 variants identified were associated with the poor metabolism of Efavirenz and potential central nervous system toxicity.” Please add a table or a figure listing those variations, and refer to it in the text.
[LINE 253] This observation was obtained by filtering the Supplementary Data files (Excel) for the corresponding gene. I have included a reference to the Supplementary data file in question)
- rs139818840” please describe the variation, here or in the discussion…. Is this table 3?
[LINE 255] Following the inclusion of the complete dataset as originally intended and the resulting influx of significant data points, variant rs139818840 was still identified but not discussed in lieu of more actionable and translational variants identified.
- ”to highlight and exclude known admixed populations such as African-American populations”. Why did you exclude the African American population? Are they not included in the American data? Please add this data back into the American data and reanalyze. You can dedicate a section to elaborate on this population genetic variability, in addition to including it in the American data section.
[LINE 363] Including these populations which are known to be admixed could potentially overrepresent some variant frequencies and underrepresent others, introducing bias, unless steps are taken to mitigate and trace the bias and resolve it. African-American populations show a high degree of admixture, making them an inappropriate proxy representation of the African or American super population's genetic diversity but rather a result of both through inheritance. This representation also becomes much more complex and nuanced when considering African-American populations on a sub-population level. Given the focus of this study on African populations with respect to historically well-studied populations used as a medical basis for technologies like ChipSeq rapid genotyping chip technologies, including African-American populations would not lend itself to this aim. For the purpose of future studies of this nature, admixture would have to be conducted to identify and control for admixture-linked over or under-representation if any, which has been noted as a study limitation at this time.
Reviewer 2 Report
Dear authors,
Please find the following comments:
# Manuscript
- Do you consider this manuscript as an article, systematic review, or meta-anaylsis study?
Please discuss the type of your manuscript with the authors.
#Abstract:
- the aim is clear but the conclusion may need some modifications and describe the aim of the study.
# Introduction:
- line 85: authors add a subtitle "Treatment and dosage"
Adding this subtitle was not appropriate since this manuscript was submitted as an article and not as as systematic review or meta-anaylsis study.
# Conclusion:
- starting from line 362: authors could add a subtitle such as "Limitations of the study" and add this part because the conclusion part is too long .
Regards.
Author Response
Dear Reviewer,
Many thanks for your revisions. Please find below a list of the changes made in relation to your comments, labelled as received:
A note on manuscript formatting issues:
When reviewing the provided comments, it was found that the majority of the provided revisions pertaining to formatting guidelines were a result of Microsoft Word formatting having gone awry and suspected issues of the manuscript during transit. To rectify this, the manuscript has since been rendered into a LaTeX document using the approved MDPI LaTeX document template.
- [MANUSCRIPT] This article is considered a manuscript.
- [ABSTRACT] As part of the major revisions required internally, the conclusion has since been updated and has been expanded upon.
- [INTRODUCTION] "Treatment and Dosage" has been corrected to "Treatment and Dosage Guidelines" to better reflect the subject matter being discussed.
- [CONCLUSION] The title and section "Limitations of the study" has been added.